# Impacts of and survival adaptations to the COVID-19 pandemic among the hill tribe population of northern Thailand: A qualitative study

**Soontaree Suratana**[1], **Ratipark Tamornpark**[1,2], **Tawatchai Apidechkul**[1,2]*,
**Peeradone Srichan**[1,2], **Thanatchaporn Mulikaburt**[1], **Pilasinee Wongnuch**[1,2],
**Siwarak Kitchanapaibul**[1,2], **Fartima Yeemard**[2], **Anusorn Udplong**[1]

**1** School of Health Sciences, Mae Fah Luang University, Chiang Rai, Thailand, **2** Center of Excellence for Hill Tribe Health Research, Mae Fah Luang University, Chiang Rai, Thailand

\* Tawatchai.api@mfu.ac.th

**Data Availability Statement:** All relevant data are within the paper and its Supporting Information files.

## Abstract

### Background

COVID-19 has exerted a variety of impacts on people, particularly people with limited education living in poor economic settings. This study investigates the impacts of and adaptations to COVID-19 among the hill tribe people of northern Thailand.

### Methods

A qualitative method was used to elicit information from key informants who lived in a hill tribe village in Mae Fah Laung district, Chiang Rai Province, Thailand. Fourteen questions on two issues were used to gather information, and an NVivo program was used to extract the findings.

### Results

A total of 57 hill tribe villagers participated, including 36 females and 21 males (mean age of 50.1 years, min = 20 and max = 90). Twenty-seven individuals were Thai Yai, 14 were Yunan Chinese, eight were Akha, and eight were members of other minor tribes. Regarding education and occupation, 30 individuals were illiterate, while 27 had attended different levels of primary school; 40 individuals were unemployed, 13 were employed as daily wage workers and farmers, and the other 4 were attending school. Three age categories were used to assess the impacts of the COVID-19 pandemic: impact of access to the educational system among the young, loss of jobs and family financial problems among the working, and access to medical care for the elderly. Six adaptation stages in response to the COVID-19 crisis were observed among the hill tribe people: shock stage with no prior experience, looking for help from health and other agencies, considering the national lockdown policy, complying with prevention and control measures, reducing stressful situations and following the new normal approach, and addressing suffering points at home and elsewhere.

**Funding:** Funding was provided by the Center of Excellence for the Hill tribe Health Research (No.01-64), Mae Fah Luang University, Thailand. The grant founders have no role and were not involved in any step of this research.

**Competing interests:** The authors have declared that no competing interests exist.

## Conclusions

The COVID-19 pandemic has exerted different impacts on different age categories among the hill tribe population living in remote and border areas. Effective adaptations have been implemented to address the new normal life under the disease, and six adaptation stages have been identified that have helped them survive the greatest threat to humankind today.

## Introduction

Since late 2019, COVID-19 has emerged globally as a new, emergent, communicable disease for humans [1]. A large number of infections and deaths have been reported throughout the year-long pandemic [2,3]. Collaborations between people and governments to address this common threat have been observed worldwide [4]. A number of impacts have been reported through forums in both formal and informal settings [5,6]. All of human society has been impacted by this disease, which can be observed in the direct and indirect effects of the disease [7–9]. There are many direct impacts, such as health problems, complications from pathogenesis, economic issues related to access to health care, and reduced quality of life [10–12]. The indirect impact can be measured in the decrease in social relationships, family relationships, quality of education due to online classes, economic losses from the lockdown, etc. [13–15]. However, the severity of the impacts depends on how people can adapt their behaviors, as well as on their education and economic status [16].

Adaptation is defined as the ability of a person or species to survive in a particular ecological niche or a behavior introduced via natural selection [17]. There are at least two forms of adaptation: biological and cultural [18]. Different populations in many scenarios have survived the pandemic [19–21]. For example, to maintain the economy, many people have begun using online markets [22]. Online teaching has been offered by educational institutes [23]. Many social and economic activities are occurring online, including business meetings [24]. Thus, the ability of different people to adapt is critical to overcoming this serious threat and reflects the maintenance capacity of populations. Those who have a high economic and education status are assumed to have a better ability to adapt until a disease-free state is achieved. However, many people live under poor economic conditions where education is also poor, including many without certain state citizenship, such as the hill tribe people in Thailand.

The COVID-19 pandemic will exert a greater impact on vulnerable populations, such as hill tribes and stateless populations, due to their economic and educational status. The hill tribes live along the border areas of Thailand and Myanmar [25]. More than 4 million hill tribe people lived in Thailand in 2020, and 30% of hill tribe people do not have Thai citizenship [26]. Those who hold Thai citizenship are able to access all public services, including health and educational institutes, free of charge [27]. The remaining 30% who are not Thai citizens live without support from the government of Thailand [27]. During the COVID-19 pandemic, the central government of Thailand launched many programs to help sustain the economic and physical health of individuals and families [28,29]. However, these programs are not provided to those without Thai citizenship [30]. Many government policies include direct and indirect measures to reduce the impacts of the disease. However, other policies have impacted village dwellers in different ways. Different age categories experience different levels and forms of impacts; moreover, individual adaptability is also related to different personal backgrounds and social support [31].

This study aimed to investigate the impacts of and adaptations to the COVID-19 pandemic and government interventions among hill tribe people living along the border of Thailand and Myanmar.

## Materials and methods

A qualitative method was applied to collect information from key hill tribe informants who were aged 20 years and over and lived in the hill tribe villages along the border areas of Thailand and Myanmar in Mae Fah Laung district, Chiang Rai Province, Thailand. A purposive method was used to select participants who had direct experience with COVID-19.

The research question guidelines were developed from a literature review and were designed to collect preliminary information from the hill tribe villagers. The questions focused on two main issues: COVID-19 impacts and adaptations. In the first set of questions, seven questions were used to determine the impacts of the COVID-19 pandemic: a) Have you been impacted by COVID-19? b) Have you had any health impacts from the disease pandemic? c) Have you had any economic impacts from the disease pandemic? d) Have you been impacted by government control and prevention measures? e) Have you experienced relationship problems from the distancing policy and measures? f) Have you experienced disparities in support from the government based on being impacted by the disease? g) Has your children's school attendance been impacted by the pandemic?.

The second set comprised seven questions used to collect information on adaptations to the COVID-19 pandemic: a) Have you implemented any self-care adaptations to prevent the disease? b) Have you implemented any adaptations to protect your family members? c) How do you modify your daily practice regarding attending social activities? d) How have you maintained your relationship with family members under social distancing? e) How have you maintained your family financial status, and how have you maintained medical access in case a family member needs to access medical care regularly? f) How have you adapted to government control and prevention measures? And g) How has your community adapted to prevent and control the disease among community members?.

All the questions were then assessed for quality by three experts in the field: social workers, public health professionals, and epidemiologists. Afterwards, questions were provided to five hill tribe people who lived in the hill tribe village in Mae Chan district, Chiang Rai Province, Thailand. Before use in the field, the question guidelines were reassessed by researchers.

The five interviewers had various backgrounds, such as medical anthropology, public health, behavioral science, and social epidemiology. Moreover, all interviewers were trained in qualitative research and had an average of four years of experience in performing qualitative research. Interviewers worked in educational institutions near the participants' residences and are familiar with one another from previous works.

Seven hill tribe village headmen were contacted after access to the villages was granted by the district government office. On the day of meeting with the village headmen, all essential information on the study was explained. Key informants were selected and informed of the appointment five days in advance. On the date of the interview, informed consent was obtained voluntarily from all participants. Those who could not speak Thai were helped by village health volunteers who were fluent in both Thai and their local languages. Village health volunteers worked as translators to improve the accuracy of the conversation between interviewer and interviewee, but the interviewer (researcher) was the one who gathered information from the participants using the question guidelines. All the participants were provided with information on the interviewers, especially their educational and occupational background. A face-to-face interview was conducted at the participants' homes, each lasting approximately 45 minutes. With the participants' approval, all interviews were taped, and field notes were taken. The interviews were conducted in January 2021.

All taped interviews were transcribed into text. The transcriptions were checked for errors before being sent back to participants to confirm the accuracy of the content. Data were

coded, and code trees were developed. The files were transferred to the NVivo program (NVivo, qualitative data analysis software; QSR International Pty Ltd., version 11, 2015). The thematic method was used to extract and organize the findings. All key findings were reconsidered by researchers before being sent to participants for their feedback. Finally, researchers drew conclusions based on all information obtained.

The research proposal, protocol, and tools were approved by the Human Research Ethical Consideration Committee from the Chiang Rai Public Health Provincial with IRB No. CRPPHO No. 72/2563. Written informed consent was obtained in Thai before the interviews were conducted. Those who did not understand the content written in Thai were provided explanations of all the content on the consent form by village health volunteers before fingerprinting into the informed consent form. All the data and records were kept confidentially and, after completion of the analysis, were destroyed properly.

## Results

Fifty-seven (57) participants provided information for the study, including 36 females and 21 males. The average age was 50.1 years, with a minimum age of 20 and a maximum age of 90. The majority were married (36), previously married (13), or single (8). Twenty-seven (27) individuals were Thai Yai, 14 were Yunan Chinese, eight were Akha, and eight were members of other minor tribes. Regarding education and occupation, 30 individuals were illiterate, 27 had attended different levels of primary school, 40 were unemployed, 13 were employed as daily wage workers and farmers, and four were attending school (Table 1).

Two sets of findings are presented in this study, one on COVID-19 impacts and the other on adaptations to the pandemic.

### A. Impacts

Several forms of impacts were detected among the hill tribe people who lived along the border area of Thailand-Myanmar. The impacts of the COVID-19 pandemic and the implementation of prevention and control measures under the regulation of the Ministry of Public Health and the central government were revealed in different aspects, such as fear of the disease, interference with lifestyle, interference with culture-related practices, and family economic shortcomings. The impacts were classified into different age categories.

**Impact among young people.**   Young people were impacted based on access to the educational system. All schools and universities have operated online to follow the guidelines required by the Ministry of Education. Many barriers were observed regarding access to education by the hill tribe children, such as internet availability and parental support in incurring costs for internet access and obtaining materials. Moreover, almost all parents were unable to help their children optimize their learning process due to a lack of previous experience and the family financial situation.

A 20-year-old man stated the following [P#3]:

"I am now studying in my first year at university. Currently, the university has announced that online classes will be implemented for the whole semester. I have many problems, including unstable internet signals, expenses for computer materials, and long but not interactive online classes. I want to get back to the classroom to meet my friends and my professors. I do not think this is good for me."

A 33-year-old woman stated the following [P#6]:

 

**Table 1. Characteristics of participants.**

| Characteristic | n | % |
|---|---|---|
| **Total** | **57** | **100.0** |
| **Gender** | | |
| Male | 21 | 36.8 |
| Female | 36 | 63.2 |
| **Age** | | |
| Mean age = 50.1, min = 20, max = 90 | | |
| **Marital status** | | |
| Single | 8 | 14.0 |
| Married | 36 | 63.2 |
| Ever married | 13 | 22.8 |
| **Education** | | |
| No education | 30 | 52.6 |
| Primary school | 6 | 10.5 |
| High school | 20 | 35.1 |
| Vocational school | 1 | 1.8 |
| **Tribe** | | |
| Tai Yai | 27 | 47.4 |
| Yunnan-Chinese | 14 | 24.6 |
| Akha | 8 | 14.0 |
| Lisu | 3 | 5.3 |
| Lahu | 4 | 7.0 |
| Yao | 1 | 1.7 |
| **Religion** | | |
| Buddhist | 48 | 84.2 |
| Christian | 9 | 15.8 |
| **Occupation** | | |
| Unemployed | 40 | 70.2 |
| Daily wage job | 13 | 22.8 |
| Attending school | 4 | 7.0 |

"I have two kids who are attending the same school. In the last semester, the school director told us the school needed to operate the class online because the new regulations from the Ministry of Education said that all schools in the whole country had to operate all classes through an online system. This led to much suffering for me and my husband because we did not have any materials to support our kids in attending their classes online. I never attended a school in my life, and I am not in the position to help my kids. We hope that the school will stop running classes online and return to the normal system."

Most young people have experienced impacts to their school- and class-related activities, particularly the need for distance learning materials.

**Impacts among those of working age.** People falling into the working-age category have faced many impacts, such as job losses, inability to visit beloved parents, inability to attend social activities, required involvement in prevention and control measures within the community, and family financial problems. Some people did not receive support from the government due to their lack of Thai citizenship. The process of applying for compensation from the government requires a smartphone and internet access, which was not possible for some hill tribe

people. Social activities that are commonly used to maintain relationships among villagers have also been prohibited by regulations.

A 55-year-old woman stated the following [P#8]:

"I lost my job in Bangkok, where I had worked for more than 10 years. My boss told me that he had to stop running the system due to the government's shutdown policy. A week later, I came back to my village. Since then, I have had no job. I have had to start farming a small piece of land to grow vegetables for survival. I have three kids and two parents that I have to support. I have not received support from the government, and I heard that very few people in the village have received government support, which included a small amount of compensation from the lockdown policy. I am surprised that the government is requiring [us] to apply for compensation through the online system because people in this village cannot do that. We do not have smartphones, and we are not familiar with filling out questions in online forms. This is also a big problem for us."

A 42-year-old man stated the following [P#31]:

"Actually, I have worked as a motorcycle taxi driver to take people to their place with a limited fee. Since COVID-19, no visitors have visited us at the village. I automatically lost my job and had no money to support my family. I have no Thai ID card, I am not a Thai citizen, but I have stayed in this village for 10 years. Thus far, I have not been compensated from the government. I have had to reduce my family's expenses. I have also started tending a small area to grow crops for the survival of my family members. However, do you know, I have been assigned many tasks by the village headman to contribute to the COVID-19 prevention and control measures."

A 31-year-old woman stated the following [P#10]:

"My parents live in other villages in another district in Chiang Rai Province. I truly want to see them, but I cannot. My parents are very old and cannot use a phone. This is a big problem to me. I cannot visit them even if we have a special event. I do not know if I will see my parents before the lockdown is over."

A 67-year-old man stated the following [P#50]:

"Do you know, during the past few months, the situation in our village is similar to being in a war. We have experienced a war between the Myanmar army and soldiers from some tribes living in the border area. At that time, nobody was moving, and nobody was talking, everyone kept quiet in their safe zone at home. The same situation returned during the COVID-19 season. Currently, I cannot go the temple to pray, which I have practiced my whole life. A few days ago, I went to the temple, and I did everything I wanted to do in a short time. Everyone in the village, we do the same thing. We cannot have interactions or talk with others for a long time."

A 29-year-old woman stated the following [P#2]:

"After the lockdown policy was implemented, I was asked to postpone my wedding ceremony in August 2020. Since then, I am still looking for my happy time with my husband. I hope that we will have a ceremony in the next few months."

This information reflects the disruption of family relationships among people who have a role in maintaining family relations and the downward trend of family finances.

**Impacts among the elderly population.** Elderly hill tribe members experience many health problems caused by physical and mental degeneration with increasing age. A large proportion of the elderly need medical care, particularly regular health checks and medicines. With the restrictions caused by many COVID-19 prevention and control measures, particularly in hospitals, accessing medical service is different from that under normal conditions. Elderly hill tribe members face unusual schedules for meeting with their doctors, which could impact their health in the future, particularly for those with noncommunicable chronic diseases.

A 65-year-old woman stated the following [P#18]:

"I have been diagnosed with diabetes and hypertension for more than 10 years and am required to check my status every month. After COVID-19, I made appointments every three months. I am not happy about this, but I have no better choice. Many times, I have had appointments to get my regular medicine at the health-promoting hospital, which is not a big hospital. I know that access to a large hospital or district hospital is dangerous for COVID-19 infections, and I need to follow more steps before meeting my doctor."

A 73-year-old man stated the following [P#11]:

"I have had diabetes problems for a while and need to see a medical doctor to maintain my blood sugar. So far, I have had appointments to see a medical doctor every month; however, last year, I only had 3 monthly appointments. I also have to take care of one of my sons who has had disability problems since birth. He cannot speak and cannot move. Last year, he was diagnosed with a bedsore that required wound dressing every day. Today, I have to do that job because a nurse told me that it is not possible to receive help at a hospital."

The major problem encountered among elderly people was difficulties accessing medical care, especially for those who need to meet with a doctor regularly.

## B. Adaptations

The hill tribe people presented adaptability to the COVID-19 pandemic in six stages: shock stage with no prior experience, looking for help from health and other agencies, considering the national lockdown policy, complying with prevention and control measures, decreasing stressful situations and following the new normal approach, and addressing suffering at home and elsewhere.

**Stage one: Shock stage.** In the first stage, after hearing about the COVID-19 pandemic, the hill tribe people reported that they were in the shock period and had no prior experience with the disease. Many people preferred to remain at home, while others tried to maintain their health by using traditional herbs and following prevention and control measures according to government guidelines.

A 78-year-old man stated the following [P#51]:

"I was very shocked by the news about COVID-19. I asked many people what the characteristics of the disease were, and everyone told me that it was not visible, which led me to experience a shock syndrome. Everyone told me that we need to keep ourselves at home. Since then, I do not want to go outside my home."

He added, "I take a shower with herbs and wash my mouth with herbs every day. I do use a mask whenever I need to go outside my home."

A 74-year-old individual stated the following [P#24]:

"I heard that COVID-19 is not visible, and I feel that it is very dangerous. Do you know, I never experienced anything like this in my whole life. I follow prevention measures, such as using masks, washing hands, and regularly taking showers."

In summary, within this period, the hill tribe people showed no specific pattern in their behaviors. However, people intended to follow the prevention and control measures that had been introduced.

**Stage two: Looking for help from health and other agencies.** Many village members looked for help, particularly from doctors and health professionals at hospitals. While the suggestions to prevent and protect in the early stage were not clear, three main prevention measures have been widely introduced: hand washing, face mask use, and social distancing. Many villagers had no concrete information about the disease, and no one could help explain to them the exact characteristics of the disease.

A 40-year-old woman stated the following [P#35]:

"I am a public health volunteer in my community. When news of COVID-19 was first heard, I asked a doctor at the community hospital how to prevent the disease. The doctor told me that no information was available. I felt that this was a big problem."

A 48-year-old man stated the following [P#29]:

"I am a village headman. In the very early days of the news regarding the COVID-19 pandemic, I received many questions from my village members, particularly how to avoid the disease and how to protect ourselves. I have asked the elderly people in the village if anyone had any experience like the pandemic of COVID-19; unfortunately, no one had experienced a similar event in the past."

In this stage, people tried to find further information from health care professionals; unfortunately, most of the validated and accurate information relevant to the disease was not available.

**Stage three: Considering the national lockdown policy.** A few months after the initial reports of COVID-19 in Thailand, the Thai central government announced a lockdown policy that did not allow anyone to move, which was intended mainly to stop contact or interactions among people. This policy led to family financial problems. Almost all community leaders were assigned to cooperate with the implementations to stop people's movements within their village.

A 33-year-old man stated the following [P#53]:

"I am a health volunteer in my community. After the lockdown was implemented, I was assigned to work on many things. We were scheduled to work in a temporary screening station or checkpoint. We worked the whole day and overnight at the station. I think we did a great job to protect our community members as well."

A 29-year-old woman stated the following [P#33]:

"When Thailand initially went on lockdown, I worked in Phuket Province as a guide for tourists from China. I earned approximately 70,000 Thai baht per month for my salary. Months later, after no visitors, I decided to move back to my village. It so sad. I am now trying to work for a small income to support my three kids in their education. My wife and I are trying to reduce family expenses to avoid unnecessary items. We started to plant vegetables in a small plot of land beside our house for daily vegetables to reduce expenses."

After the lockdown policy was implemented, almost all people followed the prevention and control measures and stayed at home. However, many people were starting to face family financial problems.

**Stage four: Complying with prevention and control measures.**   Some time after the official report on COVID-19 in Thailand was issued, many interventions were introduced. Information was released through several communication platforms, such as public television, online applications, Facebook, websites, and communication among community members. In this stage, people received saturated and simplified information about the disease, including prevention and control measures.

A 74-year-old man stated the following [P#57]:

"I could not speak or understand Thai. I have obtained information on COVID-19 from my friend who could understand Thai and speak Chinese. I also heard people talking about COVID-19 in the market. However, I am very sure that following the instructions of a doctor or health professional will save us from the disease. I regularly use a face mask and wash my hands."

A 65-year-old woman stated the following [P#18]:

"I cannot speak Thai, and I speak only Akha. I follow all measures told to me by my sons. They told me to use a face mask and regularly wash my hands. I have maintained distance from people, particularly those I do not know. I also do not invite anyone to my home."

A 33-year-old woman stated the following [P#6]:

"I have two children (8 years and 11 years) who are attending the same primary school. I ask them to wear face masks every day, I provide them an alcohol gel, and I ask them to take a shower once getting back to home. I think it is very important to build new health practices to keep the family safe."

In this stage, with no better options, all people followed the instructions suggested by the government, especially wearing face masks and cleansing their hands with alcohol gel.

**Stage five: Decreasing stressful situations and following the new normal approach.**
One year after the emergence of COVID-19, the hill tribe people were familiar with the new normal, and people started talking but continued to implement social distancing. Health practices were expanded to all age categories. Those who do not use a mask are not welcome in public. Many people have started talking with others about serious problems in their lives, such as having a job and providing medical care for those who have health problems.

A 56-year-old woman stated the following [P#8]:

"I just got back from visiting my doctor to get additional medicine yesterday. People have started visiting the hospital, but they are socially distanced and using a face mask. The

doctor has made appointments to see him every two to three months, and he told me that if I need any help before then, that I can see him anytime. I feel as if life has come back."

A 45-year-old woman stated the following [P#9]:

"I am working at the community market. In the past, people who came to purchase items did not use face masks, but today, every person uses face masks and is checked for temperature before entering the market area. Everyone is happy to follow the key measures, such as using a face mask, washing hands, and maintaining distancing."

In this stage, most people had adapted to the new normal lifestyle by maintaining distancing and wearing face masks.

**Stage six: Addressing suffering points from at home and elsewhere.** Today, many people in the hill tribe village are looking for work and beginning to make contact with other people, particularly their family members. Activities and interactions among villagers in the community have been modified into new normal forms and have recommenced. The whole situation is better than that in previous days.

A 30-year-old woman stated the following [P#55]:

"I just returned yesterday from visiting my parents, who are living in the next two villages. It was the first time I had visited in more than 6 months. We talked about many things, and I was very happy to see them in good health."

A 70-year-old woman stated the following [P#22]:

"One of my grandchildren will have her wedding ceremony next week, and we will join the event. I feel very happy. However, I will keep using face masks and maintain distancing with people."

A 20-year-old woman stated the following [P#17]:

"The university will officially start offline teaching or on-campus teaching next month. I am very excited to see friends and my teachers."

Finally, people are looking for work and joining community activities. People have become active again in all sectors, even though the conditions are not the same as they were before the pandemic.

## Discussion

A total of 57 hill tribe participants living in remote and border areas in northern Thailand described the impacts of the COVID-19 pandemic and of prevention and control measures on their lives. These impacts are discussed based on different age categories. Children attending school experienced impacts associated with obtaining effective learning due to internet access and quality and financial problems associated with obtaining the relevant materials. Job loss and family financial problems were detected in the working-age group. Several concerns about access to health care services were commonly found among the elderly. Although the hill tribe people were constrained in their ability to resolve problems, they presented a number of positive adaptations for surviving the pandemic that were detected in six stages.

In our study, the hill tribe children were impacted by the COVID-19 pandemic through a reduction in their educational effectiveness because most of the educational platforms were changed to online methods. With limited internet access and relevant materials for their education, the children were hindered in understanding the content provided through online classes. This finding is consistent with a study conducted in 62 countries in 2020, which found that students were significantly less satisfied with their academic work and life during the crisis; moreover, the education of female, full-time and first-level students and students facing financial problems was generally affected to a greater degree by the pandemic [32]. Garbe et al. [33] reported that during the COVID-19 pandemic, parents encountered several problems associated with online classes, especially learner motivation, accessibility, and learning outcomes. In Thailand, several approaches have been applied to the Thai educational system, with most teaching provided by online methods and no better options available [34]. The predominance of online teaching methods impacted a large number of students in all grades and levels, particularly in their ability to access the internet and use the associated applications [35].

In the hill tribe people of working age, many had lost their jobs during the COVID-19 pandemic, which led to family financial crises. The World Bank reported negative projections of the gross world product (GWP) and gross domestic product (GDP) in all countries in 2020 [36]. A study conducted in 77 countries on the economic impact of COVID-19 reported that countries' economies were impacted not only by COVID-19 but also by the implementation of government interventions during the COVID-19 pandemic [37]. A study in the United States reported that minorities were much more economically impacted by the COVID-19 pandemic and government interventions [38]. The Asia Foundation reported that almost all Thai people faced financial problems during the COVID-19 pandemic, particularly those who were living in poor economic and education conditions [39]. Most employees working in the industrial sector experienced the greatest impacts from the pandemic [40], and these included a large proportion of the hill tribes, who were working as employees in industrial sectors in large cities in Thailand.

The hill tribe elderly faced problems associated with their health and access to health care services during the COVID-19 pandemic. A large proportion were worried about their health and access to medicine. Bambra et al. [41] reported that under lockdown interventions to prevent and control COVID-19, elderly populations in most countries had limited access to medical care. Fischer et al. also reported that during the COVID-19 crisis, with its high numbers of infections, elderly individuals who had chronic diseases received less attention from health professionals [42]. A study on mental health problems among individuals from 78 countries found that they were common among elderly people during the COVID-19 pandemic [43].

Elderly Thai people, including the hill tribe elderly population, have been impacted by physical and mental health issues during the COVID-19 pandemic, including difficulty in maintaining doctor appointments [44].

The lockdown policy has also impacted people's relationships with their relatives and friends, which is not normally observed. This problem was compounded among vulnerable groups of hill tribe people, such as elderly people, disabled people, and children who need care from parents. In some stages, the parents were confined to a particular place or area, and movement was not allowed, which led to suffering from a lack of support. This finding is consistent with a study conducted by members of the Family Health in Europe-Research in Nursing (FAME-RN), which reported that the COVID-19 pandemic interfered with family structures and roles and influenced the health and well-being of each family member [45]. A study in Finland on the impacts of COVID-19 and family coping strategies during lockdown found that the relationship among family members was one of the substantial issues of concern for people, and the researchers attempted to identify the best coping method for Finland's

families [46]. The International Labor Organization (ILO) [47] reported that one of the crises during the COVID-19 pandemic in Thailand was family member relationships because all Thai families maintain contact based on cultural norms, and such norms are common among hill tribe families.

Six stages of adaptation under the COVID-19 pandemic crisis among the hill tribes were detected: the shock stage with no prior experience, looking for help from health and other agencies, considering the national lockdown policy, complying with prevention and control measures, decreasing stressful situations and following the new normal approach, and addressing suffering at home and elsewhere. All adaptation stages found in the hill tribe people were related to cultural and behavioral adaptations and influenced by internal and external factors. Internal factors included fear and shock from the situation, while external factors were found in a number of forms, including prevention and control strategies introduced by health and central government agencies. High adaptive ability is usually found among hill tribe people in Thailand because of their history of migration from South China to northern Thailand over two centuries. Individuals must adapt to the situations and surroundings associated with Thai culture, particularly while contacting Thai people through the trading system and Thai government officers when obtaining official documents. The hill tribe people in Thailand have high adaptability to new situations, which is clearly demonstrated by their survival under serious threats. Presti et al. [48] reported that humans adapted to survive under threats from the COVID-19 pandemic through different forms of coping and responding to certain problems. Pogrebna et al. [49] reported that people adapted their hygiene behaviors, particularly washing their hands, to protect themselves from the infection, and this represented one of the cultural adaptations made by participants in this study, as well.

Moreover, some participants reported that their living environment under COVID-19 was similar to the environment during wartime. In 1982, the Hin Tak war raged between the Thai army and the Khun Sa army in the hill tribe area [50]. People aged 40 years and over living in the hill tribe villages in the study area were exposed to the Hin Tak war and remembered the village environment at that time, with nobody moving freely. Similar feelings were reported regarding the COVID-19 situation, which reflects the fear and hopelessness of the people during the pandemic.

A few limitations exist in this study. First, some participants were not fluent in Thai; thus, translators were required, which means that the translated information might not exactly match the participants' intended meaning. However, all transcribed text was sent back to the original participant to confirm the content before analysis. Another limitation is that during the interview, the interviewer and interviewee were of different genders, and the interviewee may thus not have fully expressed their intended meaning to the interviewer.

## Conclusion

Since the COVID-19 pandemic was first announced in Thailand, many disease prevention and control measures for the population have been implemented via various approaches by local, national, and international agencies. Many of these agencies have a limited ability to reach hill tribe people who live in remote locations and border areas. Children are facing problems with receiving proper education, individuals of working age are losing their jobs and encountering new family financial problems, and the elderly population has had considerable difficulty in accessing medical care and are experiencing a loss of relationships with their offspring. In this serious situation, adaptations have been found among the hill tribes to survive the pandemic, six stages of which were observed: the shock stage with no prior experience, looking for help from health and other agencies, considering the national lockdown policy, complying with

prevention and control measures, decreasing stressful situations and following the new normal approach, and addressing suffering at home and elsewhere. Essential and accurate information on COVID-19 prevention and control in the hill tribe languages should be developed and regularly distributed to the target populations. Appropriate teaching methods should be created to support children's study at home. A mobile mental health clinic should also be developed to support people suffering from mental health problems, especially the elderly population. Moreover, programs and interventions to support and sustain family finances are also recommended.

## Supporting information

**S1 Appendix. Questions guideline.**
(DOCX)

**S2 Appendix. Data file.**
(DOCX)

**S3 Appendix. Summary data.**
(XLSX)

## Acknowledgments

We would like to thank all participants for providing essential information.

## Author Contributions

**Conceptualization:** Soontaree Suratana, Ratipark Tamornpark, Tawatchai Apidechkul.

**Formal analysis:** Soontaree Suratana, Ratipark Tamornpark, Tawatchai Apidechkul, Thanatchaporn Mulikaburt, Pilasinee Wongnuch, Siwarak Kitchanapaibul, Anusorn Udplong.

**Funding acquisition:** Tawatchai Apidechkul.

**Investigation:** Soontaree Suratana, Ratipark Tamornpark, Tawatchai Apidechkul, Peeradone Srichan, Thanatchaporn Mulikaburt, Pilasinee Wongnuch, Siwarak Kitchanapaibul, Fartima Yeemard, Anusorn Udplong.

**Methodology:** Soontaree Suratana, Tawatchai Apidechkul, Pilasinee Wongnuch, Siwarak Kitchanapaibul, Anusorn Udplong.

**Project administration:** Soontaree Suratana, Ratipark Tamornpark, Thanatchaporn Mulikaburt, Fartima Yeemard.

**Supervision:** Tawatchai Apidechkul.

**Writing – original draft:** Soontaree Suratana, Ratipark Tamornpark, Tawatchai Apidechkul, Pilasinee Wongnuch, Siwarak Kitchanapaibul.

**Writing – review & editing:** Soontaree Suratana, Ratipark Tamornpark, Tawatchai Apidechkul, Peeradone Srichan, Thanatchaporn Mulikaburt, Pilasinee Wongnuch, Siwarak Kitchanapaibul, Fartima Yeemard, Anusorn Udplong.

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
