## [Decision Letter · Decision Letter 0]

26 Mar 2021

PONE-D-21-04774

Impacts and adaptations to survive under the COVID-19 epidemic among the hill tribe population of northern Thailand: A qualitative study

PLOS ONE

Dear Dr. Apidechkul,

Thank you for submitting your manuscript to PLOS ONE. After careful consideration, we feel that it has merit but does not fully meet PLOS ONE’s publication criteria as it currently stands. Therefore, we invite you to submit a revised version of the manuscript that addresses the points raised during the review process.

This is in interesting manuscript about a population that has not received much attention amid the pandemic. Strengths are that it is largely well written, there is a reasonable amount of data presented in the quotations to support the themes, and the overview and synthesis is good. And I appreciate the focus on adaptation and resilience. However, the following will help the manuscript be acceptable for publication:

The methods require more detail. In particular, more specific information is needed on participant recruitment, data collection, and data analysis (e.g., who analyzed? what methods? citations for the data analytic method are needed; how did you deal with discrepancies in the coding?). PLOS One requests that you complete and attach the COREQ checklist to identify important aspects of the methods; in fact this will assist with identifying the missing information in the Methods section; some of the points in the checklist should be described in the text of the manuscript. (Note that it is not expected that researchers will have taken all the steps or have all the information requested in the COREQ checklist; however many of these items should be addressed. Please see other qualitative articles in PLOS One for guidance on how they have used COREQ.)

In addition to the brief introductions to each section, rather than only presenting lists of quotation, please provide at least some brief overview statement at the end of each section or some explanation of at least some of the quotations integrated with the text. It is not sufficient to merely list the quotations in each section; this presumes the reader should do the work of interpreting them and linking them to the themes. It also might be helpful to provide 1 table with the participant number, gender, and age (and perhaps other relevant information if you have it). Finally, as reviewer 1 indicates, please provide brief recommendations based on your findings; understandably these should be tentative based on 1 qualitative study.

We look forward to receiving your revised manuscript.

Kind regards,

Peter A Newman, Ph.D

Academic Editor

PLOS ONE

Journal Requirements:

3. When reporting the results of qualitative research, we suggest consulting the COREQ guidelines: http://intqhc.oxfordjournals.org/content/19/6/349. In this case, please consider including more information on the number of interviewers, their training and characteristics; and on how participants were selected.

Reviewers' comments:

Reviewer's Responses to Questions

**Comments to the Author**

1. Is the manuscript technically sound, and do the data support the conclusions?

Reviewer #1: Yes

2. Has the statistical analysis been performed appropriately and rigorously? 

Reviewer #1: N/A

3. Have the authors made all data underlying the findings in their manuscript fully available?

Reviewer #1: Yes

4. Is the manuscript presented in an intelligible fashion and written in standard English?

Reviewer #1: Yes

5. Review Comments to the Author

Reviewer #1: • The authors referred to COVID19 as epidemic, since COVID-19, according to WHO, was declared a pandemic because of the unusually fast rate in which the virus spreads. Pandemic is used when it spreads over significant geographical areas and affects a large percent of the population, while the epidemic is often used broadly to describe any problem that has grown out of control. I would suggest that the authors use the word pandemic instead.

• Since the informants are hill tribe, the authors explained that ‘However, those who could not speak Thai were helped by village health volunteers who were fluent in both Thai and their local languages’. This process should be further elaborated, the authors should declare how those helpers were trained on research data collection (interview) and ethical considerations.

• Step one: Shock the situation with no prior experience should be changed in the phase that explains the phenomenon. For example, ‘Stage of shock’

• In the conclusion, it is expected to see that the authors can provide some suggestions to the authority and policy makers how to support the hill tribe people to cope with this kind of situation.

6. PLOS authors have the option to publish the peer review history of their article (what does this mean?). If published, this will include your full peer review and any attached files.

Reviewer #1: No

---

## [Author Response · Author response to Decision Letter 0]

23 Apr 2021

Response to editor and reviewer’s comments

Dear Editor,

Thank you very much for all valuable comments and suggestions. We have revised, and improved all points of concerns. We changed the word of “epidemic” to “pandemic”, improved all section of methods using the COREQ as guideline, added table 1 to present the general characteristics of participants, added a brieft introduction on ech section 

Thank you for submitting your manuscript to PLOS ONE. After careful consideration, we feel that it has merit but does not fully meet PLOS ONE’s publication criteria as it currently stands. Therefore, we invite you to submit a revised version of the manuscript that addresses the points raised during the review process.

This is in interesting manuscript about a population that has not received much attention amid the pandemic. Strengths are that it is largely well written, there is a reasonable amount of data presented in the quotations to support the themes, and the overview and synthesis is good. And I appreciate the focus on adaptation and resilience. However, the following will help the manuscript be acceptable for publication:

: Thank you so much

The methods require more detail. In particular, more specific information is needed on participant recruitment, data collection, and data analysis (e.g., who analyzed? what methods? citations for the data analytic method are needed; how did you deal with discrepancies in the coding?). PLOS One requests that you complete and attach the COREQ checklist to identify important aspects of the methods; in fact this will assist with identifying the missing information in the Methods section; some of the points in the checklist should be described in the text of the manuscript. (Note that it is not expected that researchers will have taken all the steps or have all the information requested in the COREQ checklist; however, many of these items should be addressed. Please see other qualitative articles in PLOS One for guidance on how they have used COREQ.)

: Thank you so much for the suggestion in use the COREQ as the guideline to explain the method section, it really helps in extending the context of method section. 

: We have put a lot of the information in the method section including interviewers’ background and experience, relationship development between interviewers and interviewees, methodological orientation, method of data collection, setting, data collection and data analysis, please revised detail in page 4, lines 120-124, and 130-134; page 5, lines 136-150. 

In addition to the brief introductions to each section, rather than only presenting lists of quotation, please provide at least some brief overview statement at the end of each section or some explanation of at least some of the quotations integrated with the text. It is not sufficient to merely list the quotations in each section; this presumes the reader should do the work of interpreting them and linking them to the themes. It also might be helpful to provide 1 table with the participant number, gender, and age (and perhaps other relevant information if you have it). Finally, as reviewer 1 indicates, please provide brief recommendations based on your findings; understandably these should be tentative based on 1 qualitative study.

:Thank you for the great comment

: We have added brief introduction and summary at the end of each section, please see page 7, lines 175-177, and 194-195; page 8, lines 200-204; page 9, lines 246-247, and 254-256; page 10, lines 274-275 and 285-288; page 11, lines 304-305, 312-313, and 327-329; page 12, lines 344-346, 352-354, and 360-361; and page 13, lines 381-382; page 14, lines 403-404, 408-410 and 425-426. 

: We have added table 1 to present the characteristic of participants, please see in page 5-6.

: we have added recommendations in the conclusion sections in page 18, lines 537-543.

We look forward to receiving your revised manuscript.

Kind regards,

Peter A Newman, Ph.D

Academic Editor

PLOS ONE

Journal Requirements:

 : Thank you, we have checked and followed all instruction as the journal requirements.

: Thank you, we have added information in the section. Informed consent was obtained in written form.

3. When reporting the results of qualitative research, we suggest consulting the COREQ guidelines: http://intqhc.oxfordjournals.org/content/19/6/349. In this case, please consider including more information on the number of interviewers, their training and characteristics; and on how participants were selected.

 : Thank you for the suggestion, we have followed the COREQ in whole sections in the paper. We also attach the COREQ within this submission. Please also see the COREQ attached.

In your revised cover letter, please address the following prompts; 

 : Data are fully available in appendix section.

Reviewers' comments:

Reviewer's Responses to Questions

Comments to the Author

1. Is the manuscript technically sound, and do the data support the conclusions?

Reviewer #1: Yes

2. Has the statistical analysis been performed appropriately and rigorously? 

Reviewer #1: N/A

3. Have the authors made all data underlying the findings in their manuscript fully available?

Reviewer #1: Yes

4. Is the manuscript presented in an intelligible fashion and written in standard English?

Reviewer #1: Yes

5. Review Comments to the Author

Reviewer #1: • The authors referred to COVID19 as epidemic, since COVID-19, according to WHO, was declared a pandemic because of the unusually fast rate in which the virus spreads. Pandemic is used when it spreads over significant geographical areas and affects a large percent of the population, while the epidemic is often used broadly to describe any problem that has grown out of control. I would suggest that the authors use the word pandemic instead.

: Thank you, we totally agree with you. It’s improved in whole text including the title.

• Since the informants are hill tribe, the authors explained that ‘However, those who could not speak Thai were helped by village health volunteers who were fluent in both Thai and their local languages’. This process should be further elaborated, the authors should declare how those helpers were trained on research data collection (interview) and ethical considerations.

: Thank you for the great concern. Basically, the helpers just help in translation the message along the interview. During the interview, the interviewer (researchers) were a person on considering the direction of the interview under the question guideline. 

: This information had presented during we requested the ethical consideration, but we did not get any comment from the committee. 

: However, we agree with you to provide information about the point in the process, please see page 4, lines 130-134.

• Step one: Shock the situation with no prior experience should be changed in the phase that explains the phenomenon. For example, ‘Stage of shock’

: Thank you, we agree with you and have changed accordingly in all points.

• In the conclusion, it is expected to see that the authors can provide some suggestions to the authority and policy makers how to support the hill tribe people to cope with this kind of situation.

: Thank you, we have added recommendations in the conclusion section, please see page 18, lines 537-543.

6. PLOS authors have the option to publish the peer review history of their article (what does this mean?). If published, this will include your full peer review and any attached files.

Do you want your identity to be public for this peer review? For information about this choice, including consent withdrawal, please see our Privacy Policy.

Reviewer #1: No

Thank you,

TK

---

## [Editor Report · Decision Letter 1]

5 May 2021

PONE-D-21-04774R1

Impacts and adaptations to survive under the COVID-19 pandemic among the hill tribe population of northern Thailand: A qualitative study

PLOS ONE

Dear Dr. Apidechkul,

Thank you for submitting your manuscript to PLOS ONE. After careful consideration, we feel that it has merit but does not fully meet PLOS ONE’s publication criteria as it currently stands. Therefore, we invite you to submit a revised version of the manuscript that addresses the points raised during the review process.

The authors have done a thorough and earnest job of responding to my comments and those of the reviewer. The manuscript is much improved and basically acceptable. However, the revised text includes a number of English language errors, some of which interfere with understanding crucial statements in the manuscript. These include the following (but the authors should have the entire manuscript proofread by a professional with proficiency in English before resubmitting):

Line 147: Change to: "Written informed consent was obtained in Thai before starting the interviews." 

Line 162: Two SETS OF findings are presented.....  (not "two findings")

Line 201: Many peope did not RECEIVE support from the government due to their LACK OF Thai citizenship. (And please make sure this is sufficiently explained elsewhere, that some or many "hill tribe" people do not have official Thai citizenship.) 

Line 286: Many people preferred to REMAIN at their home, while some others TRY TO MAINTAIN THEIR HEALTH BY USING traditional herbs.....

Line 354: Change "fanatical" to FINANCIAL 

Line 425-6: People BECAME active again in all SECTORS even though this is not the same as BEFORE THE PANDEMIC

Line 537: Delete "An". Essential and accurate information....

Line 542: Moreover, programS AND interventionS to support AND sustain family FINANCES ARE also recommended. 

We look forward to receiving your revised manuscript.

Kind regards,

Peter A Newman, Ph.D

Academic Editor

PLOS ONE
---

## [Author Response · Author response to Decision Letter 1]

10 May 2021

Responses to reviewer comments

Thank you for submitting your manuscript to PLOS ONE. After careful consideration, we feel that it has merit but does not fully meet PLOS ONE’s publication criteria as it currently stands. Therefore, we invite you to submit a revised version of the manuscript that addresses the points raised during the review process.

The authors should have the entire manuscript proofread by a professional with proficiency in English before resubmitting):

Line 147: Change to: "Written informed consent was obtained in Thai before starting the interviews."

: Thank you. The whole manuscript has been reedited by American Journal Experts, No A300-3794-F762-55C0-0082. 

Line 162: Two SETS OF findings are presented..... (not "two findings")

: This issue has been addressed, and the whole text has been revised by American Journal Experts, No A300-3794-F762-55C0-0082.

Line 201: Many peope did not RECEIVE support from the government due to their LACK OF Thai citizenship. (And please make sure this is sufficiently explained elsewhere, that some or many "hill tribe" people do not have official Thai citizenship.)

: This issue has been addressed, and the whole text has been revised by American Journal Experts.

: Thank you for the comment; it should be “some” in the context of the hill tribes in Thailand.

Line 286: Many people preferred to REMAIN at their home, while some others TRY TO MAINTAIN THEIR HEALTH BY USING traditional herbs.....

: Thank you so much. This sentence has been revised.

Line 354: Change "fanatical" to FINANCIAL

: Thank you. This word has been changed.

Line 425-6: People BECAME active again in all SECTORS even though this is not the same as BEFORE THE PANDEMIC

: Thank you so much. This issue has been addressed.

Line 537: Delete "An". Essential and accurate information....

: Thank you. This issue has been addressed.

Line 542: Moreover, programS AND interventionS to support AND sustain family FINANCES ARE also recommended.

: Thank you so much. This issue has been addressed.

Journal Requirements:

: Thank you. We have checked all references, and all are correct.

Best,

TK

---

## [Editor Report · Decision Letter 2]

14 May 2021

Impacts of and survival adaptations to the COVID-19 pandemic among the hill tribe populationofnorthern Thailand: A qualitative study

PONE-D-21-04774R2

Dear Dr. Apidechkul,

We’re pleased to inform you that your manuscript has been judged scientifically suitable for publication and will be formally accepted for publication once it meets all outstanding technical requirements.

Kind regards,

Peter A Newman, Ph.D

Academic Editor

PLOS ONE
---

## [Editor Report · Acceptance letter]

18 May 2021

PONE-D-21-04774R2 

Impacts of and survival adaptations to the COVID-19 pandemic among the hill tribe population of northern Thailand: A qualitative study 

Dear Dr. Apidechkul:

I'm pleased to inform you that your manuscript has been deemed suitable for publication in PLOS ONE. Congratulations! Your manuscript is now with our production department. 

Kind regards, 

on behalf of

Dr. Peter A Newman 

Academic Editor

PLOS ONE